# UniForge: A Unified Multimodal Large Model for Detecting All-Domain Forged Image

## Abstract

With the rapid development and increasing diversification of image forged techniques, existing detection methods have exposed significant limitations in addressing emerging challenges. Current forged techniques encompass traditional methods like image manipulation, text manipulation, as well as emerging ones such as deepfake and artificial intelligence generated content. However, most existing detection models are designed to detect or localize only a single type of forgery, lacking a universal solution that can handle multiple forged methods. To address this challenge, this paper proposes a unified multimodal large model for detecting all-domain forged images named UniForge. This model aims to provide a general forged detection method to discriminate the authenticity of various types of images effectively. At its core is a novel Vision-Fusion Large Language Model, which skillfully combines the powerful feature extraction capabilities of pre-trained vision models with the outstanding semantic understanding and reasoning abilities of large language models. We have conducted extensive experimental evaluations on datasets covering various forged types, including image manipulation, text manipulation, artificial intelligence generated image, and deepfake. The results demonstrate that UniForge achieves state-of-the-art performance in the detection of all forged categories. Its comprehensive performance significantly surpasses existing methods, validating the advanced nature and excellent generalization capability of our framework.

## 1 Introduction

In recent years, the rapid advancement of artificial intelligence generated content technology has made it much easier to create and edit digital content. Therefore, it has also given rise to novel forged techniques, prominently represented by Deepfake and AI-generated images. The images produced by these techniques have reached a level of visual realism that is difficult to distinguish from authentic ones, posing a significant challenge for humans. Their misuse poses a severe threat to personal reputation and social trust. For instance, fabricated evidence can undermine judicial processes, synthetic media can be weaponized for political disinformation campaigns, and realistic-looking fake profiles can facilitate sophisticated phishing and fraud schemes. Consequently, developing detection technologies that can accurately and efficiently identify various types of forged images has become an urgent and critical task in the field of information security.

Unlike traditional tampering localization tasks(Zhou et al., 2018) (Hao et al., 2021), which focus on identifying manipulated regions within an forged image, forged image detection places a greater emphasis on determining the overall authenticity of an image. This fundamental problem of holistic, binary classification (real vs. fake) is often overlooked by existing research. Current methods are predominantly designed for specific forged types, such as models focusing only on Deepfake detection or the localization of particular tampering traces. This specialization results in a fragmented landscape of detection tools, where each tool is only effective against a narrow subset of manipulations. They lack a unified framework capable of addressing a diverse range of forged techniques. When confronted with scenarios involving a mixture of multiple forged methods, the classification accuracy of existing models still fails to meet the demands of practical applications, rendering them incapable of reliably assessing the global authenticity of an image.

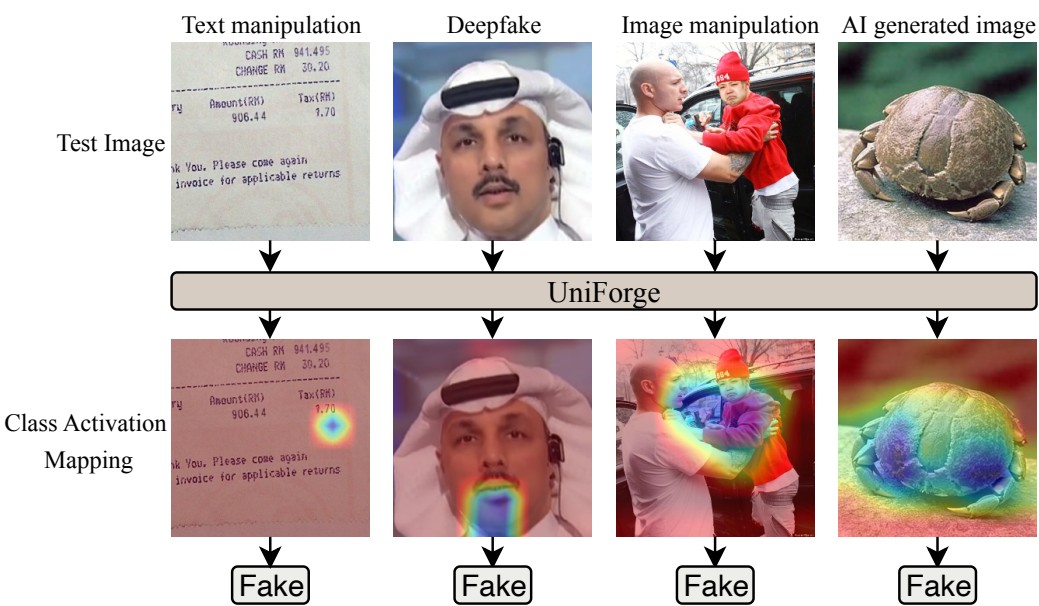

Figure 1: Grad-CAM Visualization Results of Our Method for Different Forged Images.

Furthermore, existing detection methods exhibit limitations in feature utilization. On one hand, they(Li et al., 2022) (Dang et al., 2020) primarily rely on capturing low-level, isolated visual anomalies within the image, such as subtle inconsistencies in sensor noise patterns, compression artifacts, or frequency domain irregularities. While effective for specific forgeries, these methods neglect the rich, high-level semantic context where contains forgeries. They neglect the potential of leveraging large language models for multimodal information fusion. On the other hand, even when some studies(Xu et al., 2025) incorporate multimodal approaches, they often resort to simple feature concatenation or global fusion strategies(for a detailed description, please see A.1). Such methods overlook the high-relevance requirement of features for the forged detection task, introducing a substantial amount of irrelevant and potentially confounding information during the alignment process. This, in turn, diminishes the salience of forged traces, thereby constraining detection performance.

To address the aforementioned challenges, we propose a unified multimodal large model framework named UniForge, designed to bridge the gaps in existing methodologies. This framework aims to achieve universal forged image detection across all domains using a single, unified model. UniForge leverages the powerful capabilities of a pre-trained visual encoder to extract a rich hierarchy of visual representations and a large language model (LLM) to parse textual semantics, capturing both explicit descriptions and implicit world knowledge. Through a novel prediction module, the model efficiently fuses visual anomaly features with deep-level image-text semantic inconsistencies, enabling a precise judgment of forged images. As shown in Figure1, visualization results from Class Activation Maps confirm that our model effectively focuses on the forged regions, which serves as a crucial basis for its judgment and demonstrates superior performance across various forged types.

Our main contributions are as follows:

□ 1) We present UniForge, an innovative multimodal large model framework designed to build a unified model for image forged detection. This framework is composed of three core components: a pre-trained image encoder for the extraction of deep image features; a Vision-Fusion Large Language Model (VF-LLM) that leverages these image features in conjunction with probability-guided prompts to generate a comprehensive multimodal feature; and a prediction head for forged detection that integrates both the image and multimodal features to achieve accurate predictions.

□ 2) We design a novel Vision-Fused Large Language Model (VF-LLM) for the efficient extraction and alignment of multimodal features. This model generates text embeddings highly relevant to the forged analysis task through probability-guided prompt generation. This mechanism dynami-

cally creates textual prompts that query the image's most salient and potentially suspicious aspects, guiding the LLM's reasoning process. It also introduces a multimodal query attention mechanism, which uses the text embeddings as queries to precisely retrieve and align corresponding tampering traces within the visual features. This targeted alignment ensures that only the most pertinent visual evidence is considered, effectively filtering out noise.

□ 3) Extensive experiments on multiple public benchmark datasets demonstrate that our method achieves state-of-the-art performance in detecting a wide range of forged types, including traditional image manipulation, text manipulation, AI-generated image, and deepfake. Our comprehensive evaluation shows that UniForge not only outperforms specialized detectors on their own target domains but also exhibits strong generalization capabilities to a variety of forged types. These results validate the effectiveness and superiority of our approach as a universal detection model.

## 2 RELATED WORK

**Image Manipulation Detection** With the development of deep learning, significant progress has been made in CNN-based methods for image manipulation localization. Early studies focused on capturing manipulation evidence through elaborately designed network frameworks. For instance, RGB-N(Zhou et al., 2018) uses a dual-stream architecture to process RGB content and SRM noise features separately, aiming to capture both macroscopic visual anomalies and microscopic noise disturbances, which are then fused via a bilinear pooling layer to identify inconsistencies. PSCC-Net(Liu et al., 2022a), in contrast, utilizes a bottom-up pathway to achieve coarse-to-fine localization of tampered regions through multi-scale mask refinement and cross-scale connections, effectively improving the prediction accuracy of manipulation boundaries.

To enhance detection robustness, subsequent research has explored more complex architectures. Inspired by the Vision Transformer(Dosovitskiy et al., 2021), IML-ViT(Ma et al., 2024) was the first to apply a pre-trained ViT model to this task, leveraging its powerful global context modeling capabilities while also effectively mitigating the issue of insufficient training data. Meanwhile, hybrid models that merge the advantages of CNNs and Transformers have become a research focus. For example, ObjectFormer(Wang et al., 2022a) combines the local feature extraction strength of CNNs with the global relationship modeling capability of Transformers, enabling more accurate identification of object-level semantic inconsistencies and significantly improving localization accuracy.

Detection in specific scenarios, such as document images, is also a dedicated research direction, challenged by complex backgrounds and compression artifacts. Consequently, many methods have turned to mining frequency-domain features. FFDN(Chen et al., 2024) introduces a Wavelet Pyramid Enhancement module to decompose features into high and low-frequency components, thereby enhancing the perception of weak tampering traces in the frequency domain. DTD(Qu et al., 2023) designs a frequency perception head to extract frequency cues from JPEG compression artifacts and deeply fuse them with visual features, while also adopting strategies like curriculum learning to improve model robustness by training it on examples of increasing difficulty.

**AI-generated and Deepfake Image Detection** The rapid development of generative models, including Generative Adversarial Networks and diffusion models, has led to a proliferation of highly realistic AI-generated images and Deepfakes, posing severe challenges to digital forensics. In response, the research community has developed various detection methods, targeting either general AI-generated content or the specific domain of Deepfakes.

In the field of general AI-generated image detection, particularly for diffusion models, researchers have proposed diverse strategies from different perspectives. One class of methods leverages the intrinsic properties of the generation process. For example, Dire(Wang et al., 2023) proposes distinguishing images via "Diffusion Reconstruction Error", based on the principle that the reconstruction error of a real image under a diffusion model is significantly higher than that of a generated one. Another class of methods aims to improve detection performance by designing sophisticated network architectures. DualNet(Xi et al., 2023) employs a dual-stream network to separately extract texture information and low-frequency forgery traces, while HiFiNet(Guo et al., 2023) adopts a hierarchical fine-grained approach to uniformly handle various forgery types. Furthermore, to address the generalization problem for unknown generative models, UnivFD introduces a novel strategy that utilizes the feature space of large-scale pre-trained models for detection.

In the domain of Deepfake detection, which primarily focuses on face forgery, research has also made remarkable progress. Some methods are dedicated to capturing specific forgery artifacts. For instance, Meso4(Afchar et al., 2018) is designed to capture the meso-scale properties of an image. Another major category of methods delves into clues within the frequency domain. SPSL(Liu et al., 2021a) combines spatial images with their phase spectra to capture up-sampling artifacts. F3Net(Qian et al., 2020) designs a more complex cross-attention dual-stream network to collaboratively learn frequency-aware cues for revealing anomalous features in forged images. Additionally, researchers have enhanced performance by improving network structures. For example, FFD(Dang et al., 2020) introduces an attention mechanism into the detection network to focus on tampered regions, while RECCE(Cao et al., 2022) constructs a multi-scale graph network that uses reconstruction differences as a forgery clue.

## 3 METHOD

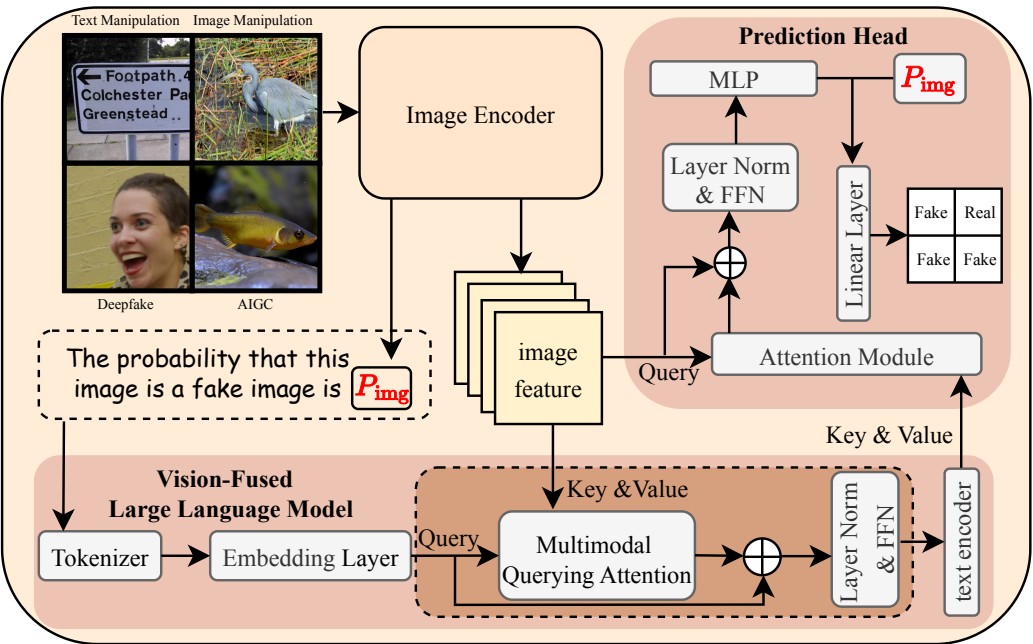

Figure 2: **Overview framework of UniForge.** The framework is primarily composed of three components: a pre-trained image encoder, a Vision-Fused Large Language Model(VF-LLM) ,and a prediction head for forgery detection.

We propose a multimodal framework for forged image detection, the core of which lies in the deep fusion of visual and textual information to achieve the precise identification of forgery traces. As illustrated in Figure 2, the overall architecture consists of three core modules: a ConvNeXt-based image encoder, a Vision-Fused Large Language Model, and an attention-based prediction head specifically designed for forgery detection.

### 3.1 OVERALL ARCHITECTURE

Given an input image $X \in \mathbb{R}^{H \times W \times 3}$, we first employ a pretrained image encoder $E_{\text{img}}$ based on the ConvNeXt architecture to extract visual features. Specifically, the encoder maps $X$ into a high-dimensional visual representation $F_{\text{img}}$ and predicts the probability $P_{\text{img}}$ that the image is a fake. To incorporate textual modality information, we construct a probability-guided text prompt: "The probability that this image is a fake image is", and append the predicted probability $P_{\text{img}}$ to the end of the prompt. The resulting combined text is then fed into our proposed Vision-Fused Large Language Model ($M_{\text{fusion}}$) to perform deeper multimodal reasoning.

$$(F_{\text{img}}, P_{\text{img}}) = E_{\text{img}}(X) \tag{1}$$

$$Y_{\text{mm}} = M_{\text{fusion}}\left(F_{\text{img}}, T \oplus P_{\text{img}}\right) \tag{2}$$

Following processing by Visual Fusion Large Language Model, we obtain a multimodal feature representation, denoted as $Y_{\text{mm}}$, which integrates both visual and textual information. These features serve as the crucial input for the subsequent forgery detection task. These features are fed into a prediction head specifically designed for image forgery detection. To fully leverage information from different modalities and achieve deep fusion, we have designed an attention-based prediction head. In this prediction head, the visual features of the image are utilized as the query, while the aforementioned multimodal feature representation $Y_{\text{mm}}$ serves as both the key and the value. This design enables the prediction head to effectively associate specific regions of the image with their corresponding multimodal contextual information.

$$P_{\text{patch}} = \text{MLP}\left(F_{\text{img}} + \text{Attention}(F_{\text{img}}, Y_{\text{mm}}, Y_{\text{mm}})\right) \tag{3}$$

$$P_{\text{final}} = \sigma\left(W_f\left[P_{\text{img}} \oplus P_{\text{patch}}\right] + b_f\right) \tag{4}$$

To ensure effective information propagation and mitigate the gradient vanishing problem, we introduce a Residual Connection after the attention module(Vaswani et al., 2017). The resulting features are then input into a Multi-Layer Perceptron to generate a patch-wise forgery probability prediction, denoted as $P_{\text{patch}}$, for each image patch. Finally, to derive a comprehensive judgment for the entire image, we integrate the global image features $F_{\text{global}}$ from the image encoder with the forgery probabilities $P_{\text{patch}}$ of the individual patches generated by the prediction head. These two information streams are fused through a final linear layer to output the final prediction, $P_{\text{final}}$, which determines whether the entire image is forged.

During the training phase, we employ the Binary Cross-Entropy loss function to optimize the network's parameters, and freeze the RoBERTa(Liu et al., 2019), the large language model used in our model. This loss function guides the model's learning process by measuring the discrepancy between the predicted probability and the ground-truth label. The formula is defined as follows:

$$\mathcal{L} = -[y \cdot \log(\sigma(P_{\text{final}})) + (1 - y) \cdot \log(1 - \sigma(P_{\text{final}}))] \tag{5}$$

Where $y$ is the ground-truth label, and $\sigma(\cdot)$ denotes the Sigmoid function. By minimizing this loss function $\mathcal{L}$, the model learns to effectively distinguish between authentic and forged images.

## 3.2 VISION-FUSED LARGE LANGUAGE MODEL

In this subsection, we will introduce the proposed Visual Fusion Large Language Model. The core of this model lies in the deep alignment and efficient fusion of textual information and visual features. First, the model takes a probability-guided text prompt as input and converts it into high-dimensional text feature embeddings through the embedding layer built into the large language model. These text embeddings not only contain rich semantic information but also provide important prior knowledge for the subsequent cross-modal fusion. To achieve deep fusion of cross-modal information, we design a multimodal query attention mechanism. Specifically, we use the aforementioned text feature embeddings as the Query, while the image features extracted from the image encoder serve as both the Key and the Value. Through this design, the text features can act as a powerful semantic prior to guide the attention mechanism to focus on the key content in the image, thereby achieving more purposeful information alignment and fusion.

Subsequently, the output of this attention mechanism, which is the visually guided weighted visual information, will be integrated with the original text embeddings through a residual connection. The introduction of a residual connection is intended to ensure that the original text guidance information is not diluted during the fusion process. This not only preserves the precise language instructions but also incorporates highly relevant visual evidence. At the same time, this structure also helps to maintain a stable gradient flow in the deep network, thereby improving the model's training

efficiency and performance. Finally, the feature vector integrated through the residual connection will be sent to the subsequent text encoder of the language model for deep contextual processing, ultimately generating a unified and information-rich multimodal feature representation.

# 4 EXPERIMENTS

## 4.1 EXPERIMENTAL SETUP

**Datasets** To achieve comprehensive model training, we construct an integrated training set spanning four key domains. We utilize the benchmark FaceForensics++(Rössler et al., 2019) dataset for the Deepfake domain, the classic CASIAv2(Dong et al., 2013) dataset for the image manipulation domain, and the large-scale GenImage(Zhu et al., 2023) dataset for the AIGC domain. For text tampering, we consolidate multiple datasets, including OSTF(Qu et al., 2025), RealTextManipulation(RTM)(Luo et al., 2024), T-SROIE(Wang et al., 2022c), and Tampered-IC13(Wang et al., 2022b). Building upon these datasets, we additionally supplement the test set with the IMD2020(Novozámský et al., 2020) and CASIAv1(Dong et al., 2013) datasets for the image manipulation domain, the DiffusionForensics(Wang et al., 2023) dataset for the AIGC domain, and the DFDC(Dolhansky et al., 2020) and Celeb-DF-v2(Li et al., 2020) datasets for the Deepfake domain.

**Implementation Details** To ensure fairness and reproducibility across all experiments, a unified training configuration is adopted for all models. The training protocol adopts the Image Forensic Fusion Protocol (IFF-Protocol)(Du et al., 2025). We employ the AdamW optimizer with hyperparameters set to $\beta_1 = 0.9$ and $\beta_2 = 0.999$, along with a weight decay of 0.05. All models are trained for 20 epochs on a single NVIDIA 4090 GPU. A cosine annealing schedule is utilized for the learning rate, which is initialized at 1e-4 and gradually decayed to 1e-5. During the data preprocessing stage, the resolution of all input images is uniformly resized to $256 \times 256$ pixels. To enhance the generalization capabilities of the models, we also apply a variety of data augmentation techniques, including random horizontal flipping, adjustments to brightness and contrast, JPEG compression, and Gaussian blur.

**State-of-the-Art Methods** For a comprehensive benchmark evaluation, we select a suite of representative methods from each domain that are both competitive and offer open-source implementations. Specifically, for the task of **Image Manipulation Detection**, we choose IML-ViT(Ma et al., 2024) and Mesorch(Zhu et al., 2025). For **Deepfake detection**, we evaluate against RECCE(Cao et al., 2022) and SPSL(Liu et al., 2021a). In the domain of **AI-Generated Content detection**, our comparison includes HiFiNet(Guo et al., 2023) and DualNet(Xi et al., 2023), and for **Text Manipulation Detection**, we select DTD(Qu et al., 2023) and FFDN(Chen et al., 2024).

## 4.2 COMPARISON WITH STATE-OF-THE-ART METHOD

Table 1: A performance comparison for image and text manipulation. The best-performing results for each test set are highlighted in bold, and the second-best values are underlined.

| Method | Image Manipulation | | Text Manipulation | | | | Avg. |
|---|---|---|---|---|---|---|---|
| | IMD2020 | CASIAv1 | OSTF | RTM | T-SROIE | Tampered-IC13 | |
| IMLVIT | 0.6844 | 0.5328 | 0.3894 | 0.2093 | 0.3401 | 0.6271 | 0.4639 |
| Mesorch | 0.7794 | 0.5672 | 0.3223 | 0.1898 | 0.2728 | 0.4796 | 0.4352 |
| HiFiNet | 0.8604 | 0.6843 | 0.3596 | 0.2472 | 0.5880 | 0.5716 | 0.5519 |
| DualNet | 0.4590 | 0.4681 | 0.3453 | 0.2065 | 0.3870 | 0.5925 | 0.4097 |
| DTD | 0.8001 | 0.6068 | 0.3656 | 0.2334 | 0.4119 | 0.5821 | 0.5000 |
| FFDN | **0.8975** | 0.6996 | 0.2935 | 0.2357 | 0.2712 | 0.5215 | 0.4865 |
| RECCE | 0.3141 | 0.3543 | 0.3458 | 0.1046 | 0.0078 | 0.5738 | 0.2834 |
| SPSL | 0.3179 | 0.6299 | 0.4437 | **0.3328** | 0.7451 | 0.7236 | 0.5322 |
| UniForge(Ours) | 0.7596 | **0.7648** | **0.7919** | 0.2116 | **0.8325** | **0.9166** | **0.7128** |

To evaluate the capability of UniForge in detecting forged images, we select several targeted datasets for each type of image forgery method. As demonstrated in Table 1 and Table 2, our proposed

method consistently outperforms other competing methods across almost all test datasets. Furthermore, given the absence of a universal method capable of detecting forgeries across all domains, we conduct a comparative analysis of UniForge against two representative methods in each respective domain.

In the task of traditional image and text manipulation detection, UniForge demonstrates exceptional performance. According to the results, UniForge achieves the best performance on four out of the six test sets: CASIAv1, OSTF, T-SROIE, and Tampered-IC13. It also secures the highest average performance, significantly outperforming the second-best method by **29.15%**. Its advantage is particularly prominent in text manipulation detection. On the OSTF dataset, its performance surpasses the runner-up by an impressive **78.4%**. Similarly, on the T-SROIE and Tampered-IC13 datasets, it outperforms the second-best results by **11.73%** and **26.67%**, respectively. Furthermore, it also obtains the top result on the CASIAv1 image manipulation dataset, outperforming the second-best by **9.32%**.

Table 2: A performance comparison for AI Generated Content and Deepfake. The best-performing results for each test set are highlighted in bold, and the second-best values are underlined.

| Method | AI Generated Content | | Deepfake | | | | Avg. |
|---|---|---|---|---|---|---|---|
| | DiffusionForensics | GenImage | FF-DF | DFDC | Celeb-DF-v2 | FF-F2F | |
| IMLVIT | 0.4345 | 0.7506 | 0.6676 | 0.6779 | **0.7939** | 0.6682 | 0.6655 |
| Mesorch | 0.6168 | 0.6665 | 0.6663 | 0.6848 | 0.7935 | 0.6668 | 0.6825 |
| HiFiNet | 0.6252 | 0.6287 | 0.6640 | 0.5843 | 0.7528 | 0.6632 | 0.6530 |
| DualNet | 0.5136 | 0.6626 | 0.6664 | 0.6477 | 0.7935 | 0.6667 | 0.6584 |
| DTD | 0.5824 | 0.6721 | 0.6661 | 0.6703 | 0.7935 | 0.6662 | 0.6751 |
| FFDN | 0.6667 | 0.6638 | 0.6664 | **0.6852** | 0.7935 | 0.6667 | 0.6904 |
| RECCE | 0.4856 | 0.8174 | 0.7162 | 0.6608 | 0.7430 | 0.7144 | 0.6896 |
| SPSL | 0.4995 | 0.9417 | 0.9150 | 0.5908 | 0.7388 | 0.8974 | 0.7639 |
| UniForge(Ours) | **0.8018** | **0.9877** | **0.9446** | 0.6482 | 0.7865 | **0.9440** | **0.8523** |

For the detection of AI-generated images and Deepfakes, UniForge also shows superior performance. According to the results, UniForge achieves the best performance on four of the six public datasets: DiffusionForensics, GenImage, FF-DF, and FF-F2F. Ultimately, UniForge ranks first among all competing methods in average performance, outperforming the second-best method, SPSL, by **11.57%**. On the AI-generated content datasets, it surpassed the runner-up by **20.26%** on DiffusionForensics and **4.88%** on GenImage. In Deepfake detection, it led the second-best methods by **3.23%** on FF-DF and **5.19%** on FF-F2F. These results indicate that UniForge is highly competitive in detecting AIGC and Deepfake content.

## 4.3 ROBUSTNESS STUDY

To comprehensively evaluate the robustness of the UniForge model, our method simulates various distortion scenarios that forged images may encounter during real-world dissemination. We employed three common image attack methods: JPEG compression, Gaussian blur, and Gaussian noise. By applying these perturbations at varying intensities to standard test datasets, we systematically examined the performance stability of UniForge in complex environments. The experimental results, as illustrated in Figure 3, indicate that the UniForge model's forgery detection performance is significantly superior to existing methods when subjected to multiple types of image attacks.

In the **JPEG compression attacks**, UniForge demonstrates outstanding robustness. On the DiffusionForensics dataset, its detection accuracy shows a notable upward trend as the compression quality decreases, showcasing a strong resilience to compression artifacts. On the FF-DF dataset, its performance remains high and peaks at the lowest quality setting. Across most datasets, including DiffusionForensics, OSTF, and FF-DF, UniForge maintains a significant performance advantage over other models, especially at more extreme compression levels.

Under **Gaussian noise attacks**, UniForge's superiority is equally evident. On the DiffusionForensics and FF-DF datasets, its performance curve is not only substantially higher than its competitors

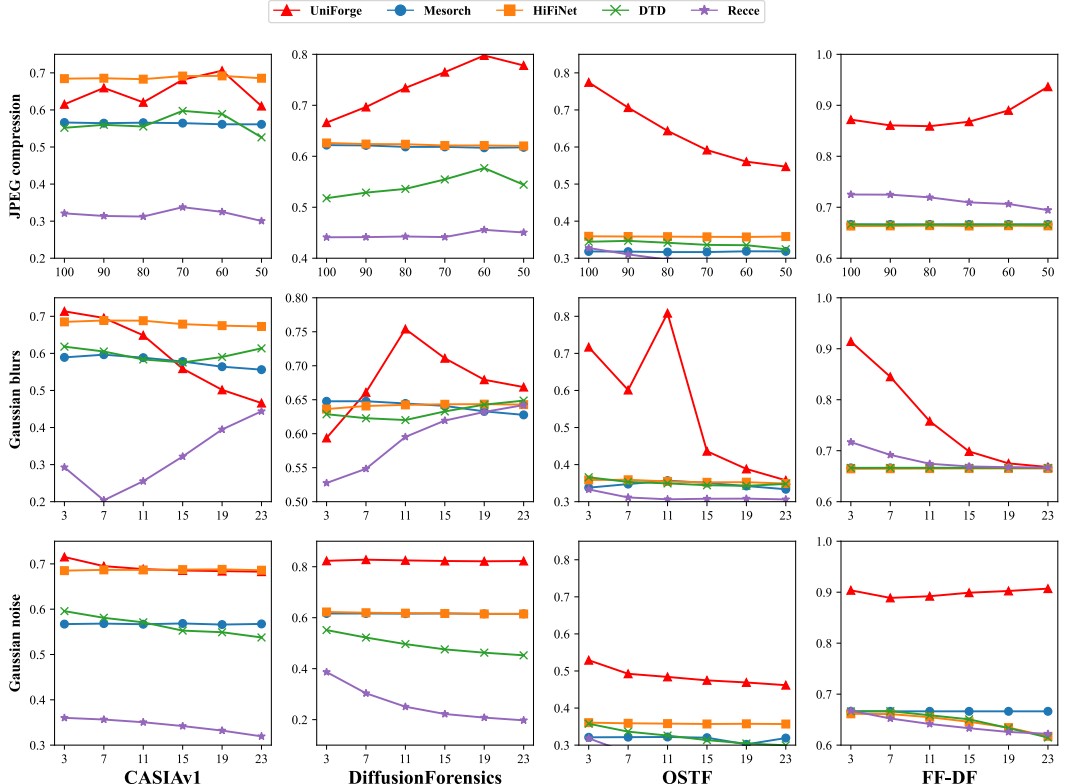

Figure 3: Robustness evaluations against three image post-processing techniques (JPEG compression, Gaussian blur, and Gaussian noise). The x-axis represents the processing intensity, and the y-axis represents the F1-score.

but also remains almost perfectly flat as noise intensity increases. This demonstrates an exceptional resilience to noise interference. On the CASIAv1 and OSTF datasets, it also consistently maintains a clear performance lead over other models.

Against **Gaussian blur attacks**, UniForge's performance is more varied but still leading in most scenarios. On datasets like DiffusionForensics and OSTF, its accuracy is volatile, yet it achieves performance peaks that are significantly higher than any other model. On the FF-DF dataset, while its accuracy trends downward with increased blurring, it consistently maintains a substantial performance margin over the other, more stable models. This indicates that while blurring impacts UniForge, it remains the most effective model overall.

We attribute the exceptional robustness of the UniForge model, particularly against compression and noise, to its unique, unified multi-modal large-model architecture. This architecture likely enables the model to fuse and leverage high-dimensional, cross-modal semantic features for decision-making. By not relying solely on low-level image forgery traces—which are easily corrupted by compression and noise—UniForge can capture more fundamental and abstract forgery cues. This results in its robust and superior performance when faced with various image post-processing attacks.

## 4.4 ABLATION STUDY

To investigate the impact of different visual backbone networks on the image feature extraction capability, we replace the original image encoder in the UniForge model, ConvNeXt(Liu et al., 2022b), with a series of mainstream vision models. These include the classic ResNet101(He et al., 2016), the self-attention-based Swin Transformer(Liu et al., 2021b), the semantic segmentation-oriented Segformer(Xie et al., 2021), the lightweight EfficientNet(Tan & Le, 2019), and Xception(Chollet,

2017), which completely utilizes depthwise separable convolutions. During this process, other modules are kept constant. The experimental results are presented in the corresponding table 3.

Table 3: Performance comparison for Image Manipulation, Text Manipulation, AI-generated image , and Deepfake datasets. The best-performing results for each test set should be highlighted in bold, and the second-best underlined.

| Model | Image Manipulation | | Text Manipulation | | AI Generated Content | | Deepfake | | Avg. |
|---|---|---|---|---|---|---|---|---|---|
| | IMD2020 | FantasticReality | OSTF | RTM | DiffusionForensics | GenImage | Celeb-DF-v2 | FF-DF | |
| ResNet101 | 0.2654 | 0.2312 | 0.2139 | 0.1176 | 0.2245 | 0.6875 | 0.7681 | 0.7998 | 0.4135 |
| SwinTransformer | 0.7196 | 0.3666 | 0.7842 | 0.2262 | 0.7464 | **0.9884** | 0.7614 | 0.9337 | 0.6908 |
| Segformer | 0.1656 | 0.1450 | 0.6901 | **0.2523** | 0.6533 | 0.9800 | 0.7475 | **0.9456** | 0.5724 |
| Xception | 0.2580 | 0.1497 | 0.5683 | 0.2068 | 0.4331 | 0.9331 | 0.5678 | 0.9317 | 0.5061 |
| EfficientNet | 0.1894 | 0.1601 | 0.1272 | 0.0230 | 0.1966 | 0.4217 | 0.7699 | 0.6512 | 0.3174 |
| ConvNext | **0.7596** | **0.5575** | **0.7919** | 0.2116 | **0.8018** | 0.9877 | **0.7865** | 0.9446 | **0.7302** |

Based on the ablation study of the image encoder, our chosen backbone, ConvNeXt, demonstrates superior performance. Its comprehensive average score across nine evaluation metrics shows an improvement of approximately **5.7%** compared to the Swin Transformer, and a significant increase of about **76.6%** over the classic ResNet101. These experiments clearly indicate that ConvNeXt possesses enhanced visual feature extraction capabilities, establishing it as a more robust and effective foundation for the overall model in comparison to other mainstream backbones.

## 4.5 FLOPs and Parameters

Table 4: Comparison of models based on Parameters and FLOPs

| Model | Params (M) | FLOPs (G) | Model | Params (M) | FLOPs (G) | Model | Params (M) | FLOPs (G) |
|---|---|---|---|---|---|---|---|---|
| Mesorch | 85.75 | 57.95 | HiFiNet | 6.89 | 145.00 | DTD | 67.07 | 272.00 |
| IML-ViT | 91.78 | 80.37 | DualNet | 7.99 | 66.34 | FFDN | 89.20 | 453.00 |
| RECCE | 25.83 | 16.18 | SPSL | 20.81 | 12.06 | **UniForge** | 229.27 | 34.36 |

Table4 presents a comparison of different models in terms of their parameter count (in millions) and floating-point operations per second (GFLOPs). As shown in the table, although our proposed UniForge model has a relatively large number of parameters (229.27M), its computational load is maintained at a low level (34.36G) through optimized calculation, which ensures a favorable inference speed. In contrast, models such as FFDN and DTD, despite having fewer parameters than UniForge, exhibit significantly higher computational complexity. This design effectively balances model accuracy and inference efficiency, ultimately achieving excellent overall performance.

## 5 Conclusion

In this paper, we propose a novel and unified multimodal large model framework, named **UniForge**, to address the limitations of current image forgery detection methods in confronting diverse and emerging forgery techniques. At the core of this framework is the Visual Fusion Large Language Model, which successfully integrates the powerful capabilities of pre-trained visual encoders with the semantic understanding of large language models, aiming to establish a universal detection solution for all forgery domains. Through an efficient multimodal feature alignment and fusion mechanism, UniForge not only captures conventional visual tampering traces but also generates text embeddings highly relevant to the forgery analysis task via probability-guided prompt generation. Extensive experiments conducted on public benchmark datasets, covering a wide range of forgery types including image manipulation, text manipulation, AI-generated image, and Deepfake, demonstrate that our model achieves state-of-the-art performance across all forgery categories. Its superior generalization ability and comprehensive performance significantly surpass existing methods, offering a robust and unified solution to the increasingly severe challenges in digital content security.

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

# A   APPENDIX

## A.1   COMPARISON WITH OTHER MULTIMODAL METHOD

In this subsection, we conduct a comprehensive comparative analysis of our proposed UniForge model and FakeShield (Xu et al., 2025). Although both employ multimodal methods to address the problem of image forgery detection, FakeShield utilizes multimodality for guidance and collaboration on the information of a single image, whereas UniForge leverages it for fusion and enhancement at the feature level. The former sacrifices some end-to-end simplicity in exchange for rich, interpretable outputs and precise localization. In contrast, the latter focuses on optimizing a single classification objective to achieve broader and more robust detection performance.

The core idea of FakeShield can be summarized as Text-guided Vision, with its architecture designed to serve the goals of interpretability and precise localization. The framework adopts a two-stage architecture, in which the Large Language Model (LLM) acts as an advanced reasoning and explanation engine. In the first stage, the LLM conducts an in-depth analysis of the input image and generates a detailed natural language description containing specifics of the tampering traces. This generated text modality is not only an interpretable output for the user but, more critically, serves as a functional "instruction" or "high-level prompt." It is then fed into the subsequent localization module to actively guide a visual segmentation model (e.g., SAM) in accurately identifying and delineating the tampered regions. It applies multimodal interaction to the decomposition and control of the image detection task, using the output of one modality to direct the operation of another. The ultimate goal is to achieve a high degree of interpretability and pixel-level localization.

In contrast, UniForge embodies the more classic and efficient concept of Multimodal Feature Fusion, with an architecture designed for all-domain, unified forgery detection. Our method constructs an end-to-end framework whose objective is not to generate complex textual explanations but to maximize the model's universal detection capabilities and robustness. Within this framework, the LLM functions as an internal feature enhancement module. It does not generate new information; instead, it deeply fuses the high-level visual features extracted by the image encoder with a text

prompt that is dynamically constructed from the model's own predicted probability. By introducing the text prompt as a semantic prior, UniForge injects high-level semantic information into the purely visual features. This enables the model to understand forgery traces from different sources from a more abstract and unified dimension. This approach focuses the multimodal technique on the enhancement and enrichment of feature representation. Through the fusion of cross-modal information, we construct a more discriminative unified representation, which in turn leads to superior performance when facing diverse types of forgeries.

## A.2 OVERVIEW OF THE DATASETS

In this section, we will introduce the datasets used in the experiment in detail.

Table 5: Summary of Our Used Datasets.

| Task | Dataset | Year | Real | Fake | Annotation |
|---|---|---|---|---|---|
| Deepfake | FaceForensics++ | 2019 | 45,388 | 127,209 | Label |
| | Celeb-DF-v2 | 2020 | 15,144 | 190,577 | Label |
| | DFDC | 2020 | 63,265 | 68,851 | Label |
| IMDL | CASIAv2 | 2013 | 7,491 | 5,123 | Label,Mask |
| | IMD2020 | 2020 | 414 | 2,010 | Label,Mask |
| | CASIAv1 | 2019 | 800 | 921 | Label,Mask |
| AIGC | DiffusionForensics | 2023 | 134,000 | 481,200 | Label |
| | GenImage | 2023 | 1,331,167 | 1,350,000 | Label |
| Document | OSTF | 2025 | 2,437 | 1,978 | Mask |
| | RealTextManipulation | 2025 | 3,000 | 6,000 | Mask |
| | T-SROIE | 2022 | 0 | 926 | Mask |
| | Tampered-IC13 | 2022 | 84 | 378 | Mask |

### A.2.1 DEEPFAKE

**FaceForensics++** Recognized as the foremost benchmark in the domain of Deepfake detection, FaceForensics++ (FF++) comprises authentic and fabricated data produced by four distinct manipulation techniques. These methods are identified as DeepFakes (FF-DF), Face2Face (FF-F2F), FaceSwap (FF-FS), and NeuralTextures (FF-NT). While each of these four subsets is generated using a different manipulation approach, they all utilize an identical set of genuine test images. The purpose of these meticulously structured subsets is to thoroughly evaluate the performance of detection models when confronted with various types of generation methods.

Regarding the composition of the dataset, the training collection is made up of 22,993 authentic images and 91,891 fabricated images. The test collection features 4,479 real images that are shared among the four manipulation subsets, with the corresponding number of fake images being 4,473 for the FF-DF subset, 4,480 for the FF-F2F subset, 4,477 for the FF-FS subset, and 4,479 for the FF-NT subset.

**DFDC** As one of the largest benchmarks for deepfake detection, the Deepfake Detection Challenge (DFDC) dataset is created by Facebook AI to accelerate research in the field. The original dataset is developed for a large-scale competition and features over 100,000 video clips with 3,426 paid actors to ensure diversity while avoiding privacy issues. DeepfakeBench(Yan et al., 2023) provides a test set with 63,265 real and 68,851 fake images, extracted from the original videos. The fabricated content is generated using several manipulation techniques, including GAN-based face-swapping, providing a comprehensive and challenging evaluation resource for detection models.

**Celeb-DF-v2** The Celeb-DF-v2 dataset is developed to address the limitations of earlier deepfake datasets, which often contain fakes with noticeable visual artifacts that make them easier to detect. This dataset provides a more challenging benchmark by featuring higher-quality, more realistic deepfakes that better reflect the current capabilities of synthesis technology. The authentic videos are

sourced from YouTube, featuring 59 celebrities with diverse ages, genders, and ethnic backgrounds, to ensure a wide range of real-world scenarios.

The dataset is comprised of 590 real videos and a significantly larger set of 5,639 corresponding deepfake videos, which are created using an improved synthesis process to minimize common flaws like color mismatch. While the core dataset consists of videos, it is often pre-processed into frames for model training and testing, with one common split resulting in 9,524 real and 179,777 fake training images, and a test set of 5,620 real and 10,800 fake images. Celeb-DF-v2 is widely recognized for its high-quality fakes and serves as a critical tool for evaluating the generalization capabilities of detection models.

### A.2.2 IMAGE MANIPULATION

**IMD2020** IMD2020 is a large-scale image manipulation detection dataset designed to overcome the diversity limitations of existing datasets. A core component of this dataset consists of 2010 authentically manipulated images collected from the internet, for which the corresponding original images and manually created binary masks of the manipulated regions are provided. Furthermore, to better simulate real-world scenarios and prevent model overfitting, IMD2020 also includes a large set of 35,000 authentic images captured by 2322 different camera models. The dataset covers various manipulation techniques, including splicing, copy-move, and forgeries generated by advanced techniques such as GANs or content-aware filling.

**CASIAv1** The CASIA Image Tampering Detection Evaluation Database Version 1.0 (CASIA V1.0) is a seminal dataset in the field of image forensics, widely recognized for its foundational role in early research on image splicing detection. The dataset comprises a total of 1,721 images, consisting of 800 authentic and 921 spliced color images. Its primary contribution lies in providing a benchmark collection of tampered images created using simple yet effective manipulation techniques, namely splicing and copy-move forgery.

**CASIAv2** CASIAv2, released by the Institute of Automation, Chinese Academy of Sciences (CASIA), is a widely recognized benchmark dataset in the domain of image tampering detection. The dataset comprises 7,491 authentic and 5,123 tampered images, covering two mainstream image manipulation techniques: splicing and copy-move. Owing to its large scale and diversity of tampered samples, CASIA v2.0 is extensively used to evaluate and compare the performance of various image tampering detection algorithms.

### A.2.3 AI-GENERATED IMAGE

**DiffusionForensics** DiffusionForensics is a dataset developed to support the assessment of detectors designed to identify images created by diffusion models. This dataset is composed of both authentic and artificially generated images from three distinct domains: LSUN-Bedroom, ImageNet, and CelebA-HQ. It features image outputs from a broad array of diffusion models, which span unconditional, class-conditional, and text-to-image generation techniques. For every image, a unique triplet of data is provided—the original source image, its reconstructed version, and the associated DIRE image—to allow for a more in-depth forensic examination. The structure of DiffusionForensics is intentionally divided into specific subsets for both training and testing purposes. By incorporating a diverse selection of generative models and image categories, it functions as a thorough benchmark for evaluating the adaptability and resilience of detectors for diffusion-based images.

**GenImage** GenImage is a massive dataset created to propel progress in the field of AI-generated image detection. It holds more than one million pairs of real and synthetic images distributed across numerous categories. The artificial images are crafted by premier generative models, such as sophisticated diffusion models and GANs. Specifically, GenImage incorporates outputs from models like ADM, BigGAN, Midjourney, VQDM, GLIDE, Stable Diffusion V1.4, Stable Diffusion V1.5, and Wukong. Each of these models contributes a nearly equal number of images (around 168,750 each), culminating in a total of 1,350,000 synthetic images. The dataset facilitates detector evaluation under practical scenarios through two distinct challenges: one is cross-generator classification, which measures a detector's ability to generalize to models it wasn't trained on, and the other is degraded image classification, which evaluates its performance against corruptions like compression, blurring, and reduced resolution. Through its combination of vast scale, diversity, and rigorous eval-

uation protocols, GenImage offers a robust benchmark for building dependable detectors for fake images.

### A.2.4 TEXT MANIPULATION

**T-SROIE** Introduced in 2022, T-SROIE stands as the pioneering dataset for identifying the locations of AIGC-type forgeries in scanned receipts through a contemporary IML methodology. This dataset features text altered by the SR-Net model and is initially distributed as large-dimension, full-frame images. We employ an identical cropping procedure to standardize all images to a $512 \times 512$ resolution and similarly crop the corresponding masks at the pixel level. Following this preprocessing, the collection designated for training is composed of 12,769 authentic and 2,747 manipulated images. The test collection holds 8,499 authentic and 1,579 manipulated images.

**RealTextManipulation** The RealTextManipulation dataset, which appears in 2025, is composed of document images that undergo both artificial and manual alterations. It encompasses a broad spectrum of forgery methods, such as copy-move, splicing, printing, and erasure, applied across various document categories like scanned forms. The source images are high-resolution and not pre-cropped. Consequently, we resize them to $512 \times 512$ and align their associated masks. After this process, the RealTextManipulation-Test set consists of 22,334 genuine and 3,444 fraudulent samples.

**OSTF** Proposed in 2025, the OSTF dataset is a collection of texts from natural scenes that are manipulated by eight distinct AIGC-based text editing models. Its primary aim is to assess a model's capacity for generalization to text manipulation techniques and image formats not previously encountered. As the source images are high-resolution and unaligned, we implement the $512 \times 512$ cropping technique, applying the identical transformation to the related masks. This results in a training set with 1,729 real and 639 fake samples, and a test set containing 14,676 real and 3,046 fake samples.

**Tampered-IC13** The Tampered-IC13 dataset, from 2022, comprises scene texts from real-world photos that are altered with the SR-Net AIGC text editing model. This dataset also does not have a predetermined cropping format, so we crop on both images and masks to achieve a $512 \times 512$ resolution. Subsequent to this preprocessing, the training partition contains 1,729 authentic and 639 counterfeit images, while the testing partition includes 1,081 authentic and 589 counterfeit images.

### A.3 SUPPLEMENTARY EXPERIMENTAL RESULTS

### A.3.1 DETAILED RESULTS OF THE GENIMAGE DATASET

This subsection provides a detailed performance of our method on the GenImage dataset. The dataset comprises images generated by eight models: ADM, BigGAN, Midjourney, VQDM, GLIDE, Stable Diffusion V1.4, Stable Diffusion V1.5, and Wukong. As presented in Table 6, we evaluated our method independently on the image subset from each model and list the respective results.

Table 6: Detailed performance comparison on the eight subsets of the GenImage dataset. The best-performing results for each test set are highlighted in bold, and the second-best values are underlined.

| Method | GenImage Subsets | | | | | | | |
|---|---|---|---|---|---|---|---|---|
| | ADM | BigGAN | GLIDE | Midjourney | Stable Diffusion v1.4 | Stable Diffusion v1.5 | VDQM | Wukong |
| IMLVIT | 0.7449 | 0.8178 | 0.7779 | 0.6588 | 0.7919 | 0.7883 | 0.6240 | 0.8012 |
| Mesorch | 0.8328 | 0.8687 | 0.8248 | 0.7464 | 0.8586 | 0.8621 | 0.6961 | 0.8494 |
| HiFiNet | 0.6214 | 0.6060 | 0.6485 | 0.6363 | 0.6589 | 0.6573 | 0.5175 | 0.6837 |
| DualNet | 0.6709 | 0.7897 | 0.7872 | 0.6406 | 0.6048 | 0.5942 | 0.5272 | 0.6857 |
| DTD | 0.6000 | 0.7010 | 0.6895 | 0.6590 | 0.7176 | 0.7141 | 0.5780 | 0.7182 |
| FFDN | 0.6458 | 0.6667 | 0.6665 | 0.6667 | 0.6662 | 0.6660 | 0.6666 | 0.6657 |
| RECCE | 0.8328 | 0.8687 | 0.8248 | 0.7465 | 0.8586 | 0.8622 | 0.6961 | 0.8494 |
| SPSL | 0.9576 | 0.9681 | 0.9519 | 0.8937 | 0.9527 | 0.9519 | 0.9214 | 0.9365 |
| UniForge(Ours) | **0.9926** | **0.9944** | **0.9899** | **0.9667** | **0.9918** | **0.9910** | **0.9932** | **0.9825** |

### A.3.2 DETAILED RESULTS OF THE ROBUSTNESS STUDY

The detailed scores of each model on the robustness tests, corresponding to the results in Figure 3, are presented in this subsection.

Table 7: The F1-scores of different models under various perturbations on the CASIAv1 dataset. For each column, the best results are in bold, and the second-best results are underlined.

| Perturbation | Model | Standard Deviations | | | | | | |
|---|---|---|---|---|---|---|---|---|
| | | 3 | 7 | 11 | 15 | 19 | 23 | Avg.F1 |
| GaussNoise | Mesorch | 0.5671 | 0.5682 | 0.5668 | 0.5685 | 0.5661 | 0.5675 | 0.5674 |
| | HifiNet | 0.6851 | 0.6868 | 0.6866 | **0.6874** | **0.6879** | **0.6860** | 0.6866 |
| | Recce | 0.3602 | 0.3565 | 0.3505 | 0.3421 | 0.3321 | 0.3194 | 0.3435 |
| | DTD | 0.5957 | 0.5810 | 0.5710 | 0.5527 | 0.5492 | 0.5376 | 0.5645 |
| | UniForge | **0.7152** | **0.6950** | **0.6885** | 0.6854 | 0.6840 | 0.6828 | **0.6918** |
| | Model | Kernel Size | | | | | | |
| | | 3 | 7 | 11 | 15 | 19 | 23 | Avg.F1 |
| GaussianBlur | Mesorch | 0.5892 | 0.5965 | 0.5885 | 0.5780 | 0.5641 | 0.5561 | 0.5787 |
| | HifiNet | 0.6850 | 0.6887 | **0.6882** | **0.6789** | **0.6748** | **0.6726** | **0.6814** |
| | Recce | 0.2926 | 0.2038 | 0.2553 | 0.3218 | 0.3947 | 0.4438 | 0.3187 |
| | DTD | 0.6182 | 0.6051 | 0.5836 | 0.5756 | 0.5903 | 0.6136 | 0.5977 |
| | UniForge | **0.7134** | **0.6956** | 0.6488 | 0.5588 | 0.5016 | 0.4660 | 0.5974 |
| | Model | Quality Factors | | | | | | |
| | | 100 | 90 | 80 | 70 | 60 | 50 | Avg.F1 |
| JpegCompression | Mesorch | 0.5661 | 0.5643 | 0.5656 | 0.5644 | 0.5613 | 0.5612 | 0.5638 |
| | HifiNet | **0.6844** | **0.6855** | **0.6830** | **0.6914** | 0.6918 | **0.6855** | **0.6869** |
| | Recce | 0.3210 | 0.3140 | 0.3124 | 0.3377 | 0.3250 | 0.3006 | 0.3185 |
| | DTD | 0.5518 | 0.5598 | 0.5553 | 0.5975 | 0.5891 | 0.5261 | 0.5633 |
| | UniForge | 0.6152 | 0.6594 | 0.6205 | 0.6814 | **0.7059** | 0.6104 | 0.6488 |

Table 8: The F1-scores of different models under various perturbations on the DiffusionForensics dataset. For each column, the best results are in bold, and the second-best results are underlined.

| Perturbation | Model | Standard Deviations | | | | | | |
|---|---|---|---|---|---|---|---|---|
| | | 3 | 7 | 11 | 15 | 19 | 23 | Avg.F1 |
| GaussNoise | Mesorch | 0.6164 | 0.6164 | 0.6158 | 0.6168 | 0.6152 | 0.6151 | 0.6159 |
| | HifiNet | 0.6224 | 0.6195 | 0.6178 | 0.6164 | 0.6147 | 0.6146 | 0.6176 |
| | Recce | 0.3868 | 0.3034 | 0.2507 | 0.2221 | 0.2080 | 0.1973 | 0.2614 |
| | DTD | 0.5514 | 0.5221 | 0.4964 | 0.4757 | 0.4627 | 0.4524 | 0.4935 |
| | UniForge | **0.8231** | **0.8276** | **0.8248** | **0.8224** | **0.8213** | **0.8223** | **0.8236** |

| | Model | Kernel Size | | | | | | |
|---|---|---|---|---|---|---|---|---|
| | | 3 | 7 | 11 | 15 | 19 | 23 | Avg.F1 |
| GaussianBlur | Mesorch | **0.6477** | 0.6478 | 0.6445 | 0.6409 | 0.6329 | 0.6277 | 0.6403 |
| | HifiNet | 0.6363 | 0.6408 | 0.6424 | 0.6433 | 0.6435 | 0.6428 | **0.6415** |
| | Recce | 0.5275 | 0.5484 | 0.5953 | 0.6193 | 0.6320 | 0.6421 | 0.5941 |
| | DTD | 0.6287 | 0.6228 | 0.6201 | 0.6328 | 0.6427 | **0.6487** | 0.6326 |
| | UniForge | 0.5937 | **0.6609** | **0.7540** | **0.7110** | **0.6794** | 0.6686 | 0.6779 |

| | Model | Quality Factors | | | | | | |
|---|---|---|---|---|---|---|---|---|
| | | 100 | 90 | 80 | 70 | 60 | 50 | Avg.F1 |
| JpegCompression | Mesorch | 0.6219 | 0.6213 | 0.6184 | 0.6184 | 0.6167 | 0.6175 | 0.6190 |
| | HifiNet | 0.6261 | 0.6240 | 0.6236 | 0.6212 | 0.6212 | 0.6203 | 0.6227 |
| | Recce | 0.4411 | 0.4414 | 0.4426 | 0.4415 | 0.4557 | 0.4505 | 0.4455 |
| | DTD | 0.5178 | 0.5287 | 0.5360 | 0.5546 | 0.5768 | 0.5442 | 0.5430 |
| | UniForge | **0.6660** | **0.6966** | **0.7341** | **0.7650** | **0.7977** | **0.7781** | **0.7396** |

Table 9: The F1-scores of different models under various perturbations on the OSTF dataset. For each column, the best results are in bold, and the second-best results are underlined.

| Perturbation | Model | Standard Deviations | | | | | | |
|---|---|---|---|---|---|---|---|---|
| | | 3 | 7 | 11 | 15 | 19 | 23 | Avg.F1 |
| GaussNoise | Mesorch | 0.3211 | 0.3215 | 0.3219 | 0.3201 | 0.3021 | 0.3191 | 0.3176 |
| | HifiNet | 0.3606 | 0.3589 | 0.3580 | 0.3570 | 0.3575 | 0.3569 | 0.3582 |
| | Recce | 0.3183 | 0.2795 | 0.2540 | 0.2386 | 0.2251 | 0.2163 | 0.2553 |
| | DTD | 0.3577 | 0.3364 | 0.3258 | 0.3135 | 0.3044 | 0.2995 | 0.3229 |
| | UniForge | **0.5290** | **0.4924** | **0.4840** | **0.4747** | **0.4690** | **0.4617** | **0.4851** |

| | Model | Kernel Size | | | | | | |
|---|---|---|---|---|---|---|---|---|
| | | 3 | 7 | 11 | 15 | 19 | 23 | Avg.F1 |
| GaussianBlur | Mesorch | 0.3378 | 0.3474 | 0.3568 | 0.3506 | 0.3422 | 0.3335 | 0.3447 |
| | HifiNet | 0.3586 | 0.3591 | 0.3545 | 0.3522 | 0.3526 | 0.3489 | 0.3543 |
| | Recce | 0.3330 | 0.3114 | 0.3064 | 0.3079 | 0.3082 | 0.3061 | 0.3122 |
| | DTD | 0.3658 | 0.3532 | 0.3493 | 0.3444 | 0.3427 | 0.3479 | 0.3506 |
| | UniForge | **0.7168** | **0.6011** | **0.8084** | **0.4361** | **0.3883** | **0.3577** | **0.5514** |

| | Model | Quality Factors | | | | | | |
|---|---|---|---|---|---|---|---|---|
| | | 100 | 90 | 80 | 70 | 60 | 50 | Avg.F1 |
| JpegCompression | Mesorch | 0.3183 | 0.3178 | 0.3166 | 0.3166 | 0.3187 | 0.3185 | 0.3178 |
| | HifiNet | 0.3591 | 0.3586 | 0.3582 | 0.3577 | 0.3574 | 0.3585 | 0.3583 |
| | Recce | 0.3276 | 0.3104 | 0.2962 | 0.2919 | 0.2754 | 0.2745 | 0.2960 |
| | DTD | 0.3445 | 0.3468 | 0.3417 | 0.3360 | 0.3351 | 0.3242 | 0.3381 |
| | UniForge | **0.7743** | **0.7063** | **0.6436** | **0.5918** | **0.5605** | **0.5470** | **0.6373** |

Table 10: The F1-scores of different models under various perturbations on the FFDF dataset. For each column, the best results are in bold, and the second-best results are underlined.

| Perturbation | Model | Standard Deviations | | | | | | |
|---|---|---|---|---|---|---|---|---|
| | | 3 | 7 | 11 | 15 | 19 | 23 | Avg.F1 |
| GaussNoise | Mesorch | 0.6663 | 0.6663 | 0.6663 | 0.6663 | 0.6663 | 0.6663 | 0.6663 |
| | HifiNet | 0.6614 | 0.6609 | 0.6548 | 0.6458 | 0.6344 | 0.6172 | 0.6458 |
| | Recce | 0.6675 | 0.6523 | 0.6415 | 0.6332 | 0.6258 | 0.6222 | 0.6404 |
| | DTD | 0.6662 | 0.6666 | 0.6586 | 0.6506 | 0.6334 | 0.6149 | 0.6484 |
| | UniForge | **0.9037** | **0.8888** | **0.8921** | **0.8990** | **0.9024** | **0.9070** | **0.8988** |
| | Model | Kernel Size | | | | | | |
| | | 3 | 7 | 11 | 15 | 19 | 23 | Avg.F1 |
| GaussianBlur | Mesorch | 0.6663 | 0.6663 | 0.6663 | 0.6663 | 0.6663 | 0.6663 | 0.6663 |
| | HifiNet | 0.6644 | 0.6647 | 0.6650 | 0.6652 | 0.6652 | 0.6651 | 0.6650 |
| | Recce | 0.7168 | 0.6918 | 0.6744 | 0.6691 | 0.6677 | 0.6667 | 0.6811 |
| | DTD | 0.6662 | 0.6663 | 0.6663 | 0.6663 | 0.6663 | 0.6663 | 0.6663 |
| | UniForge | **0.9142** | **0.8450** | **0.7579** | **0.6985** | **0.6751** | **0.6680** | **0.7598** |
| | Model | Quality Factors | | | | | | |
| | | 100 | 90 | 80 | 70 | 60 | 50 | Avg.F1 |
| JpegCompression | Mesorch | 0.6663 | 0.6663 | 0.6663 | 0.6663 | 0.6663 | 0.6663 | 0.6663 |
| | HifiNet | 0.6631 | 0.6634 | 0.6639 | 0.6632 | 0.6637 | 0.6634 | 0.6635 |
| | Recce | 0.7250 | 0.7247 | 0.7194 | 0.7096 | 0.7064 | 0.6944 | 0.7133 |
| | DTD | 0.6663 | 0.6658 | 0.6660 | 0.6662 | 0.6660 | 0.6660 | 0.6661 |
| | UniForge | **0.8719** | **0.8606** | **0.8590** | **0.8678** | **0.8900** | **0.9364** | **0.8810** |

