# OpenReview forum: "UniForge: A Unified Multimodal Large Model for Detecting All-Domain Forged Image"
_ICLR.cc/2026/Conference — ICLR 2026 Conference Withdrawn Submission_

### Official Review · Reviewer_poVK · 2025-10-18

**Soundness:** 2
**Presentation:** 1
**Contribution:** 1
**Rating:** 2
**Confidence:** 4

**Summary:**

The paper proposes UniForge, a unified multimodal large model framework for all-domain image forgery detection. Unlike prior works that are typically specialized for a single type of forgery (e.g., Deepfake, AI-generated images, or traditional image manipulation), UniForge aims to provide a universal detection approach capable of handling diverse forgery sources. The core component is a Vision-Fused Large Language Model (VF-LLM) that combines a ConvNeXt-based visual encoder with a RoBERTa-based language model through a probability-guided text prompt and multimodal query attention mechanism. This design allows the model to fuse semantic and visual evidence for authenticity judgment. Extensive experiments are conducted on multiple benchmark datasets spanning four domains — image manipulation, text manipulation, AI-generated content, and Deepfake — demonstrating that UniForge achieves state-of-the-art (SOTA) results and superior robustness under various perturbations (JPEG compression, blur, and noise). The paper also includes ablation studies on different visual backbones and complexity analysis.

**Strengths:**

- The work addresses the important and emerging challenge of detecting forged content across multiple domains, which is crucial in the era of AIGC and multimodal misinformation.

- The paper conducts extensive experiments across a wide range of datasets and forgery types, providing detailed comparisons against SOTA methods and robustness tests under post-processing attacks.

**Weaknesses:**

- The statement in the Introduction section, “This fundamental problem of holistic, binary classification (real vs. fake) is often overlooked by existing research,” seems problematic because a large number of works (e.g.,TruFor[1], FakeShield[2], SIDA[3]) have studied this problem.

- The article raises two issues with existing methods: 1) They are unable to detect tampered images in different scenarios. 2) They rely solely on low-level features or fail to fuse multimodal features effectively. However, from a methodological perspective, the authors fail to conduct a detailed analysis and design specific to issue 1 or illustrate how the proposed multimodal information fusion strategy improve the versatility of the detection model.

- Is $P_{img}$ a probability vector? How is it obtained from the Image Encoder? The shape should be clearly stated.

- In the design of the VF-LLM, the input is just a fixed text and $P_{img}$, so where does the term "rich semantic information" come from?

- IML-ViT[4] and Mesorch[5] only perform localization task, how does the detection results come from?

- What is the evaluation metric in comparison experiment?

- In Ablation Study, there should be a comparison between w/ and w/o VF-LLM.

Overall: This work does not contribute enough to the field. The article raises the question of "achieve universal forged image detection across all domains using a single, unified model", but does not provide further inspiring views or perspectives to solve this problem. As the main innovation, VF-LLM starts from the perspective of multimodal information alignment, but this approach also has similar structures in other papers (e.g., HEIE[6]), and the article lacks relevant comparisons. There is also a lack of relevant ablation experiments to prove the effectiveness of VF-LLM. The structure of the Method section is a bit confusing, and some detailed information is not clearly stated.

Minor Issues:

- In the Related Work section, it is recommended to summarize the common defects of existing methods in order to contrast them with the method proposed in the article.

 - The second to last line on page 3 shows that UnivFD did not cite any relevant literature.

- In Eq. (4), what $W_f$ and $b_f$ are should be clearly stated.

[1] TruFor: Leveraging all-round clues for trustworthy image forgery detection and localization

[2] FakeShield: Explainable Image Forgery Detection and Localization via Multi-modal Large Language Models

[3] SIDA: Social Media Image Deepfake Detection, Localization and Explanation with Large Multimodal Model

[4] IML-ViT: Benchmarking Image Manipulation Localization by Vision Transformer

[5] Mesoscopic Insights: Orchestrating Multi-Scale & Hybrid Architecture for Image Manipulation Localization

[6] HEIE: MLLM-Based Hierarchical Explainable AIGC Image Implausibility Evaluator

**Questions:**

Please refer to the weaknesses.

---

### Official Review · Reviewer_ABah · 2025-10-23

**Soundness:** 2
**Presentation:** 2
**Contribution:** 2
**Rating:** 2
**Confidence:** 5

**Summary:**

This paper proposes UniForge, a unified multimodal large model for universal detection of diverse image forgeries. The core methodological innovation is a Vision-Fusion Large Language Model, which integrates features from pre-trained vision models with the semantic reasoning capabilities of Large Language Models. This architecture is designed to provide a general solution for discriminating authenticity across various forgery types, including manipulation, deepfakes, and AIGC. Experiments show that the proposed method achieves promising results.

**Strengths:**

1, unified image forgery detection is an interesting problem and worth to study.
2, the paper is easy to follow
3, The experiment results is good and comprehensive

**Weaknesses:**

1, the proposed VF-LLM's archtecture is similar to the vision language models, encoding the image embedding and fuse them with the text via LLM, why the authors did not directly use a pre-trained vision language models like Qwen 2.5-VL as the backbone.

2,  Given that the attention and MLP are both well-known techniques, what's the technical contribution of this paper？

3， There are no special design for generalization enhancement, does the general detection ability comes from the well-trained feature encoding modules, ConvNeXt and RoBERTa?

4,  In Table 1, UniForge falls behind the competiion methods by a clear margin on IMD2020 dataset, why is it the case?

**Questions:**

see weaknesses

---

### Official Review · Reviewer_RW2E · 2025-10-27

**Soundness:** 2
**Presentation:** 3
**Contribution:** 2
**Rating:** 2
**Confidence:** 5

**Summary:**

This manuscript proposes UniForge, a unified multimodal model for detecting diverse image forgeries (image/text manipulation, AIGC, deepfake). It first extracts visual features with ConvNeXt, then uses a probability-guided Vision-Fused LLM to focus on suspicious regions, and finally applies an attention-based prediction head to fuse global and patch cues.

**Strengths:**

The manuscript is well-structured and easy to follow.

**Weaknesses:**

**Limited methodological innovation.** The core designs of the UniForge framework - a pretrained vision backbone, a frozen LLM/text encoder, and a cross-modal attention scheme that takes text as query and vision as key/value — are already standard in recent multimodal forensics systems such as FakeShield, ForgeryGPT[1], and CLIP-based universal  AIGC detectors. As a result, the work mostly repackages established multimodal components for a broader mix of forgery domains (image, document/text, AIGC, deepfake), so the methodological advance over prior works is incremental rather than fundamental.

**Performances not leading among de facto SOTA.**
The experimental comparison does not include the most recent SOTA methods. In particular, for fast-evolving areas like deepfake and AIGC detection, this work only compares with pre-2023 baselines, while stronger detectors and updated benchmarks are already available in 2025. It is recommended to add comparisons with the latest works(e.g. Effort[2]) to make the claim of superiority more convincing.

[1] ForgeryGPT: Multimodal Large Language Model For Explainable Image Forgery Detection and Localization. arXiv:2410.10238.

[2] Orthogonal Subspace Decomposition for Generalizable AI-Generated Image Detection. ICML 2025.

**Questions:**

For image manipulation and text manipulation forgeries, recent works place emphasis on pixel-level / region-level forgery localization than on image-level binary detection.
Can the proposed Vision-Fused LLM and attention-based prediction head be extended to output forgery masks? If not, does this limit the strength of the “all-domain” claim for these localization-oriented tasks?

---

### Official Review · Reviewer_BLRZ · 2025-10-29

**Soundness:** 2
**Presentation:** 2
**Contribution:** 2
**Rating:** 4
**Confidence:** 5

**Summary:**

This paper proposes UniForge, a unified multimodal framework for detecting forged images across diverse domains, including traditional manipulations, text manipulations, AI-generated content, and deepfakes. The method combines a visual encoder with a vision-fused large language model (VF-LLM) via a probability-guided prompt to jointly reason over visual and textual representations. The model outputs both patch-level and global forgery predictions. Extensive experiments on multiple benchmark datasets show that UniForge outperforms existing domain-specific methods and maintains robustness under common image perturbations. The approach aims to offer a general, all-domain solution to image forgery detection.

**Strengths:**

1. The paper addresses a significant gap by proposing a unified framework capable of handling multiple types of image forgeries (e.g., image manipulation, text-based forgery, AI-generated content, deepfakes), which are typically handled by domain-specific models.

2. The integration of visual features and a large language model (LLM) using a probability-guided prompt and cross-modal attention is novel in this context. It allows the model to leverage semantic understanding alongside visual cues for improved detection.

3. The authors conduct extensive experiments across a wide range of forgery domains and datasets. The model demonstrates strong performance and robustness under common image distortions (e.g., JPEG compression, Gaussian blur/noise), supporting its generalizability.

**Weaknesses:**

1. The overall architecture is relatively simple and follows a straightforward combination of a vision encoder and a frozen LLM module. The method lacks sufficient architectural innovation beyond integrating existing components.

2. The model does not fine-tune the LLM; instead, it freezes its weights and uses it merely as a feature extractor. However, the paper lacks detailed explanations about how the prompts are designed, how textual embeddings are obtained, and whether any prompt engineering was performed. Moreover, the decision to freeze the LLM rather than fine-tune it is not well justified—fine-tuning could potentially improve performance and deserves discussion.

3. The ablation experiments focus primarily on comparing different visual backbones while keeping other components fixed. It would be valuable to see ablations on the VF-LLM module itself—for instance, removing the probability-guided prompt, disabling multimodal attention, or using only the visual/text encoder—to better quantify the contribution of each architectural element.

4. As shown in Table 4, UniForge has significantly more parameters than most baseline methods. It remains unclear whether the performance gains mainly stem from the larger model size rather than the proposed design. A fair comparison under similar parameter budgets would strengthen the empirical claims.

5. Figure 2 is unclear and lacks alignment with the textual descriptions. Important mathematical symbols and notations referenced in the text are missing from the diagram. It is recommended to add the necessary formula symbols and legends in Figure 2 to help understand and correspond to the content of the article

**Questions:**

Please refer to the weaknesses.

---

### Note · Authors · 2025-11-15

I have read and agree with the venue's withdrawal policy on behalf of myself and my co-authors.